# Wastewater-Based Epidemiology of SARS-CoV-2 and Other Respiratory Viruses: Bibliometric Tracking of the Last Decade and Emerging Research Directions

Hassan Waseem [1] , Rameesha Abid [2], Jafar Ali [3], Claire J. Oswald [4] and Kimberley A. Gilbride [1],*

1   Department of Chemistry and Biology, Toronto Metropolitan University, Toronto, ON M5B 2K3, Canada;
    hassan.waseem@torontomu.ca
2   Department of Microbiology, Quaid-i-Azam University, Islamabad 44100, Pakistan;
    rameeshaabid@bs.qau.edu.pk
3   Key Lab of Groundwater Resources and Environment, Ministry of Education, Jilin University,
    Changchun 130021, China; jafaraliqau@gmail.com
4   Department of Geography and Environmental Studies, Toronto Metropolitan University,
    Toronto, ON M5B 2K3, Canada; coswald@torontomu.ca
*   Correspondence: gilbride@torontomu.ca

**Abstract:** The COVID-19 pandemic has prompted an overwhelming surge in research investigating different aspects of the disease and its causative agent. In this study, we aim to discern research themes and trends in the field of wastewater-based epidemiology (WBE) of SARS-CoV-2 and other respiratory viruses over the past decade. We examined 904 papers in the field authored by researchers from 87 countries. Despite the low reported incidence of COVID-19 in 2023, researchers are still interested in the application of WBE to SARS-CoV-2. Based on network visualization mapping of 189 keyword co-occurrences, method optimization, source, transmission, survival, surveillance or early-warning detection systems, and variants of concern in wastewater were found to be the topics of greatest interest among WBE researchers. A trend toward evaluations of the utility of new technologies such as digital PCR and WBE for other respiratory viruses, particularly influenza, was observed. The USA emerged as the leading country in terms of research publications, citations, and international collaborations. Additionally, *Science of the Total Environment* stood out as the journal with the highest number of publications and citations. The study highlighted areas for further research, including data normalization and biosensor-based data collection, and emphasized the need for international collaboration and standardized methodology for WBE in future research directions.

**Keywords:** bibliometrics; COVID-19; wastewater-based epidemiology; influenza; SARS-CoV-2; WBE

## 1. Introduction

The growing corpus of scientific evidence highlights the potential of wastewater-based epidemiology (WBE) as a useful approach for determining disease incidence in different populations [1,2]. WBE offers valuable insights and addresses surveillance challenges that routine clinical monitoring encounters [3]. Human excretions, comprising urine, feces, mucus, saliva, etc., are a significant anthropogenic contributor to wastewater, primarily directed into septic tanks and sewer lines [4]. These excretions, especially from infected individuals, may contain biological markers of the causative agents of various diseases, including respiratory viruses. Sewer lines are the first point in the wastewater drainage and treatment system where the detection of viruses can occur, often before the diagnosis of clinical cases [5]. Sewer systems transport the drainage water and waste matter to wastewater treatment plants (WWTPs) for treatment before releasing it into the environment. Despite the apparent usefulness of wastewater-based monitoring, it does not provide insights into individual disease severity. Nevertheless, it offers the advantage of being unbiased, as it is not influenced by variations in access to clinical testing and health-seeking behavior [6].

WBE serves as a powerful resource for predicting the propagation of virulent factors, genetic material, and pathogens, including SARS-CoV-2, within an area [7,8]. The non-invasive approach allows for testing populations at once, making it more cost-effective in comparison to traditional clinical surveillance. This approach is especially useful for detecting the etiological agent of COVID-19, where individuals with mild or no symptoms at all are present. WBE has also been successfully employed for the assessment of poliovirus, antimicrobial resistance genes (ARGs), and norovirus [9–12]. After the emergence of COVID-19, wastewater-based SARS-CoV-2 monitoring has been utilized to identify and highlight community areas with the need for diagnostic testing and to assist in addressing the problem of the scarcity of clinical testing [13,14].

Furthermore, continuous use of WBE has paved the way for considering its applicability to other respiratory viruses, such as influenza [15]. The comprehensive nature of WBE and its ability to capture a broad population spectrum make it an attractive technique for other viruses as well [16,17]. By analyzing wastewater samples, researchers can not only screen for the presence of viral RNA but also gain insights into the prevalence and genetic diversity of circulating strains within a community [18]. Such studies can serve as a valuable tool for public health authorities to make well-informed decisions regarding targeted interventions and vaccination strategies and to take disease mitigation measures. In summary, studying other respiratory viruses in wastewater, apart from SARS-CoV-2, is a proactive approach to safeguarding public health, enhancing surveillance systems, and maintaining a state of preparedness for any future respiratory virus-related challenges.

Bibliometrics is a reliable methodology for measuring growth and advancement in a specific research field. The concept of bibliometrics was originally proposed by Alan Pritchard in 1969 [19]. Recently, bibliometric analyses were utilized in several related research fields, including wastewater treatment [20–22]. A few studies also conducted an analysis of publication outputs and the performance of countries and institutions in the context of SARS-CoV-2 using bibliometrics [23,24]. In the past, scientists employed different tools for generating and analyzing bibliometric data, including Bibexcel, CiteSpace, and VOSviewer software [20,25,26]. These tools helped researchers visualize and identify various patterns in the published literature, identify influential and leading authors, research groups, and institutions, and assist in highlighting the latest and emerging trends in the specific domain.

WBE proved to be highly informative in the initial stages of the pandemic, providing crucial insights at a time when public health data were limited. Its relevance among policymakers and scientists remains significant in the present era, especially with the apparent reduction in clinical monitoring of SARS-CoV-2 in different countries [27,28]. The rising interest among policymakers, researchers, and scientists in contributing to WBE for studying SARS-CoV-2 and different RNA viruses has resulted in a sudden expansion in this field. The widespread application of WBE faces numerous challenges and uncertainties [29–33]. There remains a dearth of comprehensive bibliometric studies encompassing the future directions of WBE, especially in the post-COVID-19 scenario. To the best of the authors' understanding, no prior study has performed an extensive bibliometric analysis of WBE in relation to SARS-CoV-2 and other respiratory viruses. To address this gap, this study aims to offer a comprehensive overview of WBE's application to SARS-CoV-2 and other respiratory viruses. To achieve this objective, the study employs bibliometric analyses to explore the following research questions:

(1) How has the research on WBE in the context of SARS-CoV-2 and respiratory viruses evolved over time in terms of scale?
(2) What are the primary research themes emerging from scientific publications related to WBE and respiratory viruses?
(3) What research gaps exist, and what are the potential future research directions for WBE in the post-COVID-19 pandemic era?

Our study provided a snapshot of the literature related to the WBE of respiratory viruses, including SARS-CoV-2, using VOSviewer. To identify the research gaps in the

literature, our work assessed different parameters of WBE, including virus concentration methods and population biomarkers used for RNA normalization in wastewater. The structure of the remaining research article is as follows: Section 2 contains the methodology. Section 3 contains the bibliometric results, including a keyword co-occurrence analysis, global contributions, and institutional and authors' impact in the context of WBE of respiratory viruses, especially SARS-CoV2, which are presented and discussed. Section 4 identifies research gaps and highlights future research directions to enhance WBE practices. Finally, in Section 5, the major results are summarized and concluded.

## 2. Methodology

### 2.1. Data Acquisition and Search Strategies

The Web of Science (WOS) is a widely recognized subscription-based online platform for accessing scholarly references and citation data [34]. To gather the most relevant publications, we conducted a search on the WOS Core Collection Database to obtain the most relevant publications using specific search phrases (Table 1). A total of 916 documents, published from 2014 to 6 July 2023, pertaining to WBE in the context of SARS-CoV-2 and respiratory viruses, were retrieved. After a manual screening to exclude irrelevant papers and duplicates, a total of 904 documents were selected. The metadata records of these documents were extracted from the database, including article titles, authors, publication years, document type, source of publication, author keywords, cited reference count, countries/regions, publishers, and authors' affiliated institutes.

**Table 1.** The parameters used in the data collection procedure.

| Sr. No. | Items | Results |
| --- | --- | --- |
| 1 | Search String | ((((((TS = (Monitoring of SARS-CoV-2 and respiratory viruses in wastewater))) OR TS = ((Surveillance of SARS-CoV-2 and respiratory viruses in wastewater))) OR TS = ((Wastewater surveillance of respiratory viruses))) OR TS = ((Wastewater based epidemiology of SARS-CoV-2)))) OR TS = ((((Wastewater based epidemiology of COVID-19)))) OR TS = ((Wastewater surveillance for COVID-19)) OR TS = ((SARS-CoV-2 surveillance in wastewater)) |
| 2 | Query Link | https://www.webofscience.com/wos/woscc/summary/842163db-8860-4094-a5b6-c5e15dbc5da1-96169533/relevance/1 |
| 3 | Database | Web of Science |
| 4 | Initial number of publications retrieved | 916 |
| 5 | Search date | 6 July 2023 |
| 6 | Inclusion criteria | Only English material, data limited to 2014–2023 |
| 7 | Final number of publications retained for analysis | 904 |

### 2.2. Study Design and Data Analysis

This study mapped the research themes and evaluated future research directions for wastewater monitoring of SARS-CoV-2 and other respiratory viruses using a bibliometric approach. The bibliometric analysis was employed using the VOSviewer program, version 1.6.16, designed by van Eck and Waltman [35], to assess publications. Specifically, the performance of WBE for SARS-CoV-2 and respiratory-virus-related scholarly output was examined and mapped by depicting bibliometric metrics emphasizing (1) a keyword co-occurrence analysis, (2) the geographical coverage of the studies, (3) a document co-citation analysis (DCA), (4) institutional and authors' publications and citations, and (5) highly cited publications to identify study patterns and trends of research activity. The extracted bibliometric data were cleaned by (i) integrating the English language parameter,

(ii) amalgamating singular and plural words, and (iii) eliminating general phrases lacking a specific meaning.

VOSviewer was used to visualize the results of the density analysis, network visualization of the authors' keywords, and network visualization of DCA, which is a bibliometric technique used in the field of scientometrics to analyze relationships between scientific documents. It focuses on how frequently two documents are cited together by other documents.

## 3. Results and Discussion

### 3.1. Type and Distribution of Articles

The temporal and geographic distribution of the articles provided information about the global progression of WBE for respiratory viruses, especially SARS-CoV-2. All documents in WOS were organized into different categories, with research articles representing the greatest number (775), followed by reviews (111), editorial materials/letters (14), and proceedings/meeting abstracts (4). Only four relevant articles were retrieved from 2014 to 2019, i.e., prior to the commencement of the COVID-19 outbreak. These four articles are not shown in Figure 1 but are included in all our analyses. The sharp and sudden increase in scientific publications from 2020 to 2023 was accompanied by the rise in confirmed clinical cases of COVID-19 (Figure 1), emphasizing the importance of WBE as a strategy for community surveillance over the course of the pandemic. As per our analysis, the first research study was published in *Letters in Applied Microbiology* on 1 February 2020. The study described the presence of SAFV (Saffold virus) in Italian sewage samples [36]. Afterward, numerous reviews were also published. The first relevant review article was published on 11 July 2020. The study featured useful and important COVID-19 reference materials. Particularly in the summer of 2020, there was a significant surge in publications. A total of 59 documents were published before 15 December 2020. Additionally, until December 2021, December 2022, and 6 July 2023, more than 276, 378, and 189 were published, respectively.

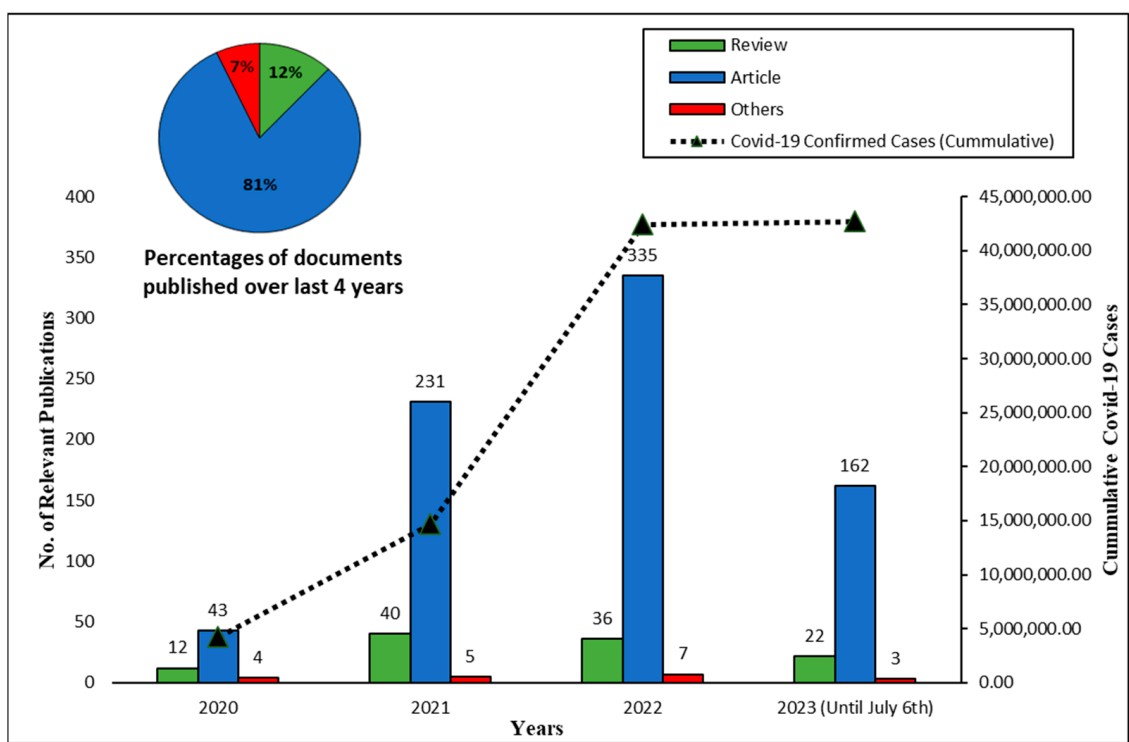

**Figure 1.** Year-wise distribution of document types along with cumulative COVID-19 confirmed cases. COVID-19 cases were retrieved from the Johns Hopkins Coronavirus Resource Center database.

*3.2. Bibliometric Tracking of the Research Trends by Keywords*

The assessment of keyword occurrence over time suggests how research themes evolved before and during the COVID-19 pandemic. After cleaning the keyword data, 189 distinct keywords were found. The network visualization map was divided into four main clusters, where each cluster represented specific areas of research within the broader field of WBE for SARS-CoV-2 and respiratory viruses (Figure 2a). For example, Cluster 1 was focused on research pertaining to method development, especially virus concentration methods and population biomarkers. Research in Cluster 2 was based mostly on sources and different parameters impacting the survival of the virus in wastewater. Cluster 3 was focused mostly on epidemiological and/or surveillance-based studies. Lastly, Cluster 4 included studies that were mostly concerned with the circulation and transmission of variants of concern (VOCs) in wastewater. The different types of keywords within the clusters also show the diversity of approaches in the analysis of viruses, the broadness of the comprehensive field of WBE, and its far-reaching impacts on wastewater systems and public health.

A density visualization heat map was then created for the same 189 keywords using VOSviewer (Figure 2b). The most popular keywords were "SARS-CoV-2" and "wastewater", which are in accordance with the searched strings. An obvious observation is the type of amplification techniques used for viral RNA detection in WBE studies. Most WBE studies carried out throughout the world employ reverse transcription polymerase chain reaction (RT-PCR) and quantitative reverse transcription polymerase chain reaction (RT-qPCR), as evident from the cumulative keyword occurrence of 164 in our analysis. However, the occurrence of droplet digital PCR (ddPCR), rt-ddPCR, and digital PCR keywords (29 times) highlighted the trend of the introduction of digital PCR use in WBE studies. One potential reason for the introduction of digital PCR trends in WBE studies could be the efficiency of traditional qPCR detection, which is also influenced by PCR inhibitors commonly found in wastewater [37]. Another major advantage of digital PCR over its conventional counterpart is the direct absolute quantification of viral RNA due to its non-reliance on traditional standard curves. Additionally, due to the higher efficiency of digital PCR at lower concentrations of DNA/RNA and because of less interference from PCR inhibitors, digital PCR is reported to have lower limits of detection (LODs) and limits of quantification (LOQs) in comparison to qPCR, particularly in WBE studies [38,39]. This is quite an important aspect, as lower viral concentrations are usually expected in wastewater owing to high dilution [40].

Variants of concern (VOCs) in wastewater are another important theme in WBE studies. The VOCs exhibit specific mutations that can potentially impact viral infectivity [41,42]. Our analysis underlines the significance of this research area, with a total of 50 keyword occurrences. Whole-genome sequencing (WGS) has been used to detect VOCs in wastewaters in different parts of the world [43–45]. Wastewater sequencing can inform researchers about the emergence of a new variant in a region before the strain is detected by sequencing the clinical samples in the same region. Similarly, the sequencing of wastewater samples can also play a part in the identification of novel genetic sequences. However, for the routine monitoring of VOCs, alternative approaches have been developed by scientists to detect VOCs [46], including allele-specific qPCR, digital PCR, nested RT-PCR assays, and mass spectrometry [47,48]. For example, Xiaoqing Xu and co-workers utilized allele-specific RT-qPCR assays to successfully differentiate eight SARS-CoV-2 variants in sewage [49]. Some research groups are also combining amplification and sequencing technologies to track the circulation of VOCs; for example, Rasmussen et al. reported the use of RT-PCR and nanopore sequencing to detect variants [50]. Such a combinatorial technique is especially useful for scenarios where known variants of concern in limited quantities are present. By utilizing these techniques, a concurrent assessment of different co-circulating viral strains and types can be performed, which, in turn, allows for the timely detection of drifted variants, subsequently prompting public health directives for testing and management. The implementation of wastewater surveillance also presents an

opportunity for the efficient surveillance of infrequent yet critical emergencies, for example, the emergence of a novel respiratory virus strain at the human–animal interface with severe potential public health impacts.

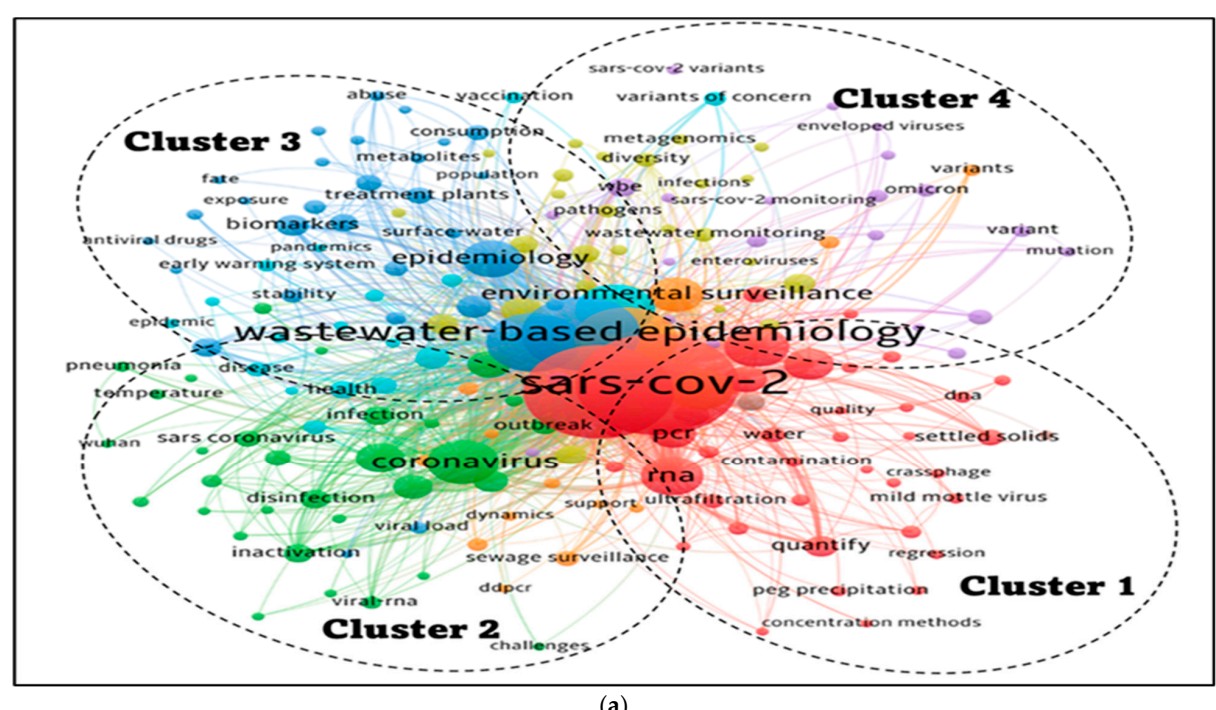

(**a**)

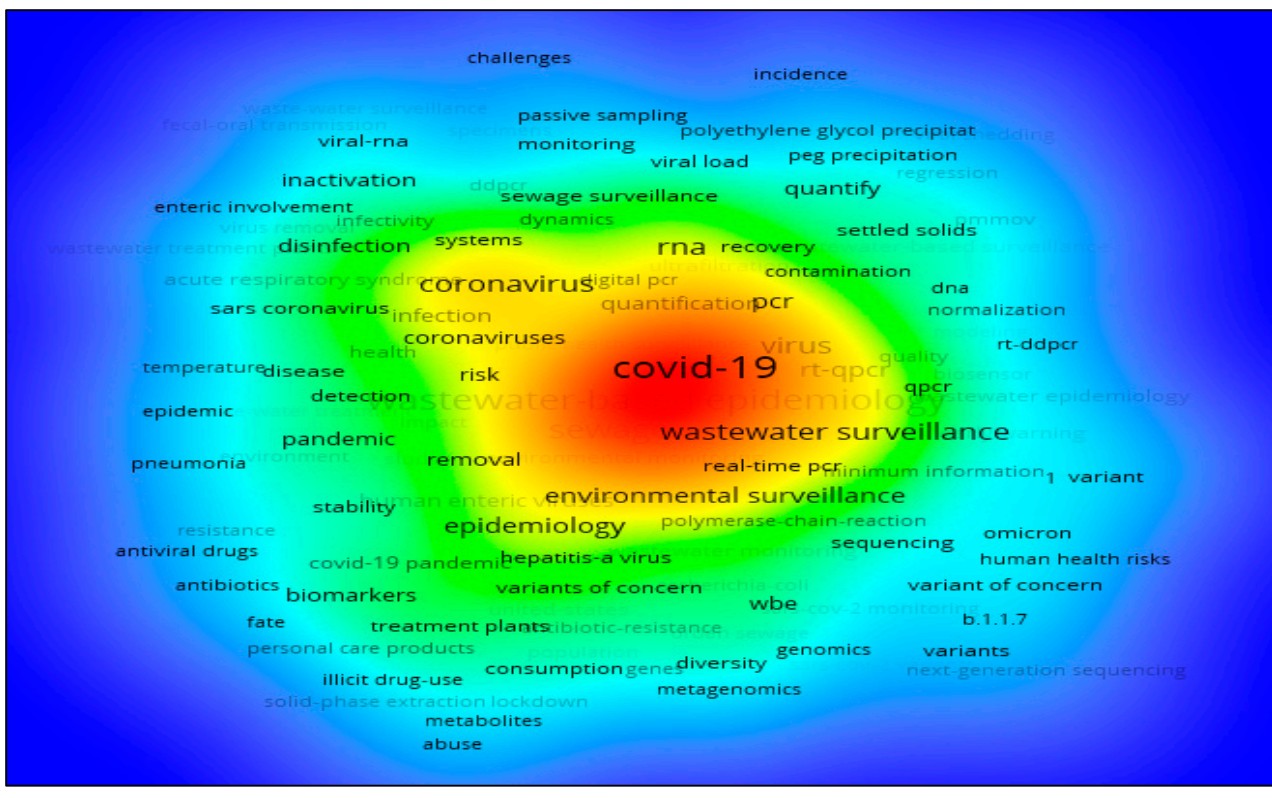

(**b**)

**Figure 2.** (**a**) Network visualization map based on keyword occurrences from 904 selected publications. (**b**) Density visualization map based on keyword occurrences from 904 selected publications. The terms "SARS-CoV-2" and "wastewater" were excluded from the density plot to reduce the bias of general terms.

Wastewater is a complex matrix with variable biological, chemical, and physical characteristics that can impact the stability of RNA and its detectability. The complexity of the wastewater mixture ultimately leads to spatiotemporal variability in WBE results and uncertainty in viral concentrations, particularly at low levels. To address these issues, scientists often perform pre-treatment steps, including the utilization of virus concentration methods, to mitigate the impact of PCR inhibitors. Our keyword analysis revealed that scientists are actively investigating the issues of the basic complexities of wastewater and viral dilutions by using several sample pre-treatment and virus concentration steps [51–53]. Inter-laboratory comparisons for identifying the best concentration principles were also published [54]. Upon an in-depth manual analysis, we noted that most of the studies utilized a smaller wastewater volume (<250 mL); however, quite a few studies reported that an increased wastewater sample volume can enhance the overall sensitivity of WBE and that the use of ultracentrifugation techniques for virus concentration is apparently the most common one [55,56]. However, our keyword analysis found only polyethylene glycol (PEG) precipitation and ultrafiltration as the most commonly occurring keywords, with 14 and 13 occurrences, respectively. The keyword "adsorption" occurred ten times, representing the use of membrane adsorption, another important virus concentration method. Recently, a few studies have been published comparing different virus concentration methods in WBE. For example, Zheng et al. reported that ultracentrifugation outperformed all the other virus concentration methods [57]. However, the choice of concentration method is highly dependent on equipment availability, labor intensity, cost-effectiveness, and throughput considerations.

WBE data normalization is a process of correcting measurements to account for variations or uncertainties due to dilution of fecal material by stormwater, groundwater infiltration, or other solid material sources (e.g., leaf litter). The Center for Disease Control and Prevention (CDC) recommends normalizing viral wastewater concentrations before calculating trends to consider waste input, dilution, and other factors. Normalizing viral concentrations in wastewater based on the amount of human feces is crucial for interpreting and comparing viral levels over time. Pepper mild mottle virus (PMMoV), a commonly found RNA virus in human fecal material, was utilized for WBE data normalization due to its higher concentrations in wastewater [58]. It can be quantified simultaneously with targeted respiratory viruses using multiplexed assays. PMMoV, also recommended by the US CDC, is widely recognized as the most popular fecal biomarker. In the keyword search analysis, occurrences of PMMoV and its alternative names were significantly higher than those of crAssphage and other biomarkers. However, some studies argue in favor of other markers for normalization. For instance, Shu-Yu Hsu et al. concluded that the chemical marker PARA serves as a more reliable population marker compared to PMMoV [59]. In yet another study, PMMoV normalization was not able to improve correlations between WBE data and clinical cases at most sites [60]. Fluctuations in municipal sewage input and normalizing viral concentrations with wastewater flow were also recommended [61,62]. Researchers recently investigated the impact of viral flow, population density, and biomarker standardization on correlation and detection time. The findings indicated that funding flow meters for small WWTPs could be more beneficial compared to PMMoV testing [60]. In summary, although PMMoV normalization is the most abundantly used normalization technique, no consensus is present on which WBS normalizing parameter leads to the strongest correlations between WBE data and clinical cases. More research in this direction would be helpful for the standardization of WBE data normalization.

Among other respiratory viruses, the scientific community has been mostly interested in applying WBE to the influenza virus, which occurred 13 times as a keyword. In fact, influenza was the only respiratory virus that was identified in our keyword search after SARS-CoV-2. After the start of the COVID-19 pandemic, most of the research was focused on the wastewater-based evaluation of SARS-CoV-2. However, recently, the trend of assessing the dynamics of other respiratory viruses, particularly the influenza virus, has started surfacing [63–65]. Researchers have recognized an immediate requirement for

enhanced surveillance of influenza due to the significant impacts of seasonal and pandemic strains on global public health. This urgency stems from the disruption of traditional patterns of seasonal occurrence of respiratory viruses, probably owing to enforced and voluntary behavioral changes, e.g., usage of masks, sanitizers, restricted mobility, etc., that respond to atypical respiratory disease dynamics. Thus, an imperative need to re-evaluate and improve surveillance mechanisms for influenza has been realized. The occurrence of non-respiratory viruses, e.g., human enteric viruses [24] and hepatitis A virus [19], was found to be greater than the influenza virus [13], but we will not be discussing them as they are out of the scope of our article.

*3.3. Global Contributions to WBE Research on Respiratory Viruses*

In order to be included in the analysis, a country, organization, or author must have a minimum of five documents, as per the defined threshold value. Based on the WOS-retrieved data, the documents span 87 countries/regions (out of which 51 met the threshold) related to this field, involving 1747 institutions (out of which 139 met the threshold) and 5581 authors (out of which 167 met the threshold) globally. One of the key markers of a country's interest in a particular domain is the number of published papers in that country. The country distribution of articles highlighted the dominance of the USA, Australia, India, England, and Canada. Researchers from these five states contributed 656 publications, whereas the USA alone contributed more than one-third of the total papers. The 363 articles from the USA covered different domains of WBE, including wastewater surveillance [66], virus detection and quantification [67], epidemiological correlations [68], seasonal variations and outbreak predictions [69], pandemic preparedness [70], public health strategies [71], genomic characterization and comparative analysis [72], and the reemergence of respiratory viruses [64]. Although WBE can be a highly useful tool globally, most of the WBE research has been reported only from high- and middle-income countries (Figure 3). The total number of studies undertaken in developing countries tends to be limited. One possible explanation for this is that the consistent monitoring or surveillance of pathogens via WBE is associated with expensive infrastructure and higher costs of reagents and chemicals, making it out of the reach of most low-income countries. The disruptions caused by the pandemic, reflecting the impact of SARS-CoV-2, were evident. Various measures taken by different countries, including travel restrictions, the closure of academic institutions, and closed borders, impeded communications across all spheres of life. However, as evident from Figure 3, international cooperation and research efforts remained resilient, particularly at the academic interface. The countries with the most publications were usually the ones with the most collaborations, except for Canada, which falls out of the top ten countries in terms of collaboration links. A report of an advisory panel, comprising academic leaders, on the federal research support system also highlighted challenges, especially in establishing international partnerships for Canadian researchers [73]. Additionally, the Canadian government's security policies may also be a barrier to international research collaborations with major research hubs, such as China [74].

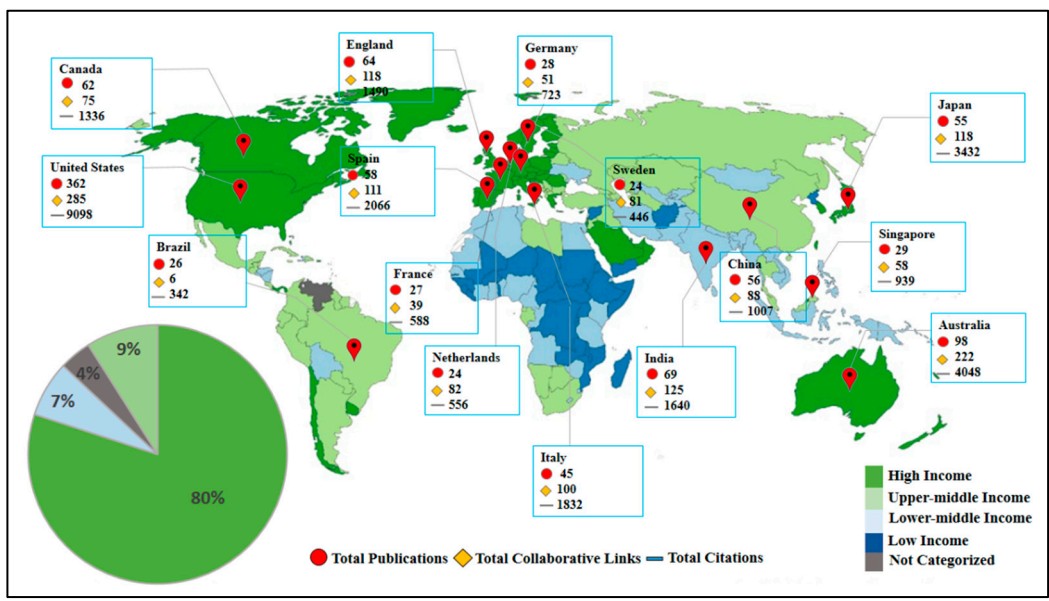

**Figure 3.** World map showing the top 15 countries/regions based on articles published on the WBE of SARS-CoV-2 and respiratory viruses. The colors of the maps show the World Bank country classification by income level.

### 3.4. Document Co-Citation Analysis

Document co-citation analysis (DCA) is one of the most frequently used approaches in bibliometric research. It focuses on how frequently two documents are cited together by other documents. Figure 4 represents a document co-citation network analysis. A total of 21,704 cited references were explored. We only selected references that had more than 20 citations and obtained a total of 342, which were subsequently utilized for the mapping of the co-citation network.

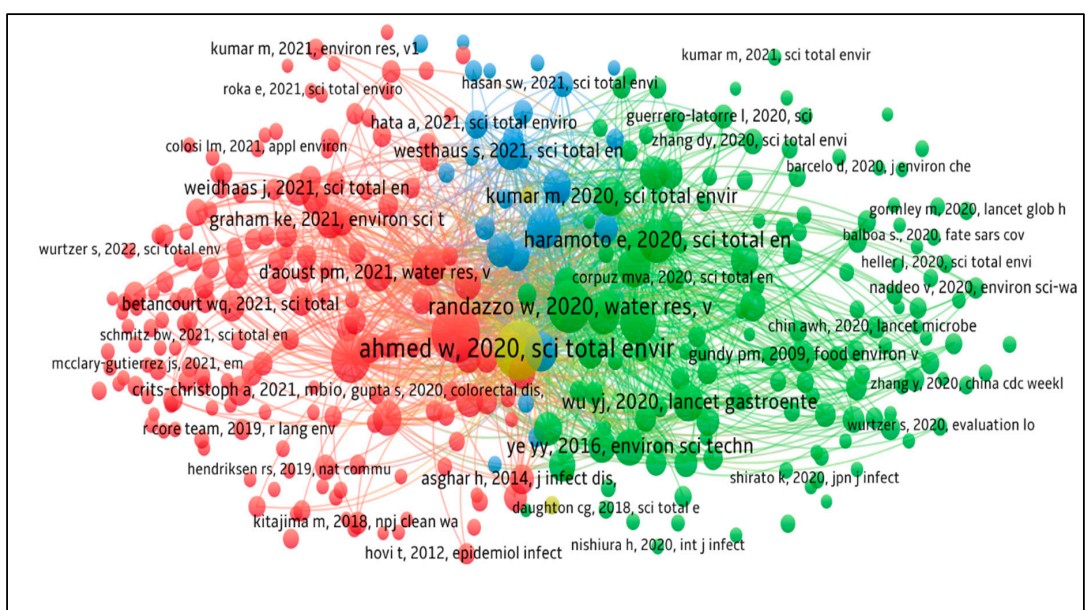

**Figure 4.** Network visualization map based on highly co-cited documents from 904 selected publications. The research clusters are represented by colors.

The clusters in the co-citation network represent knowledge structures, and the majority of the references belong to three clusters, i.e., red, green, and blue. A review of the top highly co-cited articles in each structure was performed. Information about the top five

highly co-cited papers in each cluster can be found in Supplementary Table S1. In the red cluster, highly co-cited documents belonging to the initial reports and developments of the WBE of SARS-CoV-2 in different countries were present. Whereas, in the green cluster, the references were focused on improvements in sampling sites/strategies and method development to support public health decisions. Lastly, in the red cluster, the references were concerned with sequencing analyses for deciphering variants of concern.

### 3.5. Institutional and Author-Level Contributions

Contributions at institutional and individual levels play a vital and multifaceted role in research across all disciplines. Similarly, their role in the development of the WBE of SARS-CoV-2 and respiratory viruses cannot be neglected. Our study also collected the fundamental details from the literature regarding the top ten most significant institutes (Table 2). With 32 documents published, Hokkaido University had the greatest number of publications worldwide. The University of Notre Dame, Australia, ranked number 2 on the list, followed by the CSIRO Land and Water Institute. Among the top ten institutions, the USA and Australia had three institutions, Japan had two institutes, and the United Kingdom and Singapore had one institution. Regarding the frequently cited authors of the relevant publications, Kitajima Masaaki, currently working as an Assistant Professor at the Division of Environmental Engineering of Hokkaido University since 2016, was the most highly cited author with 2844 citations, followed by Ahmed Warish, a Principal Research Scientist associated with CSIRO in different research roles for the last 13 years. However, based on total publications, Ahmed Warish was the leading author, with a total number of publications of 32 in comparison to Masaki, who had 29 published articles (Table 3).

**Table 2.** Top most relevant institutions/organizations in publishing WBE of SARS-CoV-2 and respiratory viruses.

| Ranks | Institutes | Country | Documents | Citations | Citation per Publication |
|-------|-----------|---------|-----------|-----------|--------------------------|
| 1 | Hokkaido University | Japan | 32 | 2884 | 90.1 |
| 2 | University of Notre Dame | Australia | 29 | 3022 | 104.2 |
| 3 | CSIRO Land and Water Institute | Australia | 26 | 2657 | 102.1 |
| 4 | Stanford University | United States | 22 | 610 | 27.7 |
| 5 | University of Yamanashi | Japan | 21 | 1732 | 82.47 |
| 6 | Arizona State University | United States | 20 | 901 | 45.05 |
| 7 | Bangor University | United Kingdom | 20 | 784 | 39.2 |
| 8 | The University of Queensland | Australia | 20 | 1829 | 91.45 |
| 9 | MIT | United States | 19 | 625 | 32.89 |
| 10 | Nanyang Technological University | Singapore | 19 | 585 | 30.78 |

**Table 3.** Authors with the highest no. of publications and citations in WBE of SARS-CoV-2 and respiratory viruses.

| Rank | Authors | Total Documents | Total Citations | Citation per Publication |
|------|---------|-----------------|-----------------|--------------------------|
| 1 | Ahmed, Warish | 32 | 2633 | 82.2 |
| 2 | Kitajima, Masaaki | 29 | 2844 | 98 |
| 3 | Bivins, Aaron | 28 | 2317 | 82.7 |
| 4 | Bibby, Kyle | 21 | 2600 | 123.8 |
| 5 | Kumar, Manish | 20 | 615 | 30.7 |
| 6 | Simpson, Stuart l. | 19 | 1826 | 96.1 |
| 7 | Haramoto, Eiji | 18 | 1433 | 79.6 |
| 8 | Farkas, Kata | 14 | 573 | 40.9 |
| 9 | Jiang, Guangming | 14 | 323 | 23.0 |
| 10 | Joehm, Alexandria b. | 13 | 327 | 25.1 |

*3.6. Impact of Articles and Journals*

The frequency of citations is one method for assessing an article's impact on a related field of study. Table 4 shows the top ten frequently referenced articles in the domain of WBE and SARS-CoV-2. We know that sewer overflow can cause serious public health concerns worldwide, and the pre-existing challenges have been worsened by the presence of SARS-CoV-2 in wastewater. Sojobi and Zayed published the article "Impact of Sewer Overflow on Public Health: A comprehensive scientometric analysis and systematic review" in the *Environmental Research Journal* on 30 June 2021, which has been cited 479 times by the cut-off date of our study. Using scientific methods and a thorough literature review, the authors tried to build an association between sewer overflow and public health, particularly COVID-19 [75]. Based on Table 4, there is an increased interest in evaluating the occurrence, fate, and analysis of viruses in wastewater based on other highly cited articles [76–78]. In addition, method optimization for WBE is a field that has received immense attention [79]. One of the top cited papers identified in our bibliometric analysis was focused on the opportunities and challenges for biosensors and nanoscale analytical tools, which further highlights the importance and trend of research in this unique direction [80]. Similarly, Mehdi Nourinejad and colleagues suggested the implementation of a sewage manhole sensor system that would enable an accurate, robust, and cost-efficient method for the real-time assessment of SARS-CoV-2 within communities [81]. If developed, such a technique can simplify the instruments required for WBE and may result in significant cost savings while also serving as a basis for action.

One of the most significant and popular platforms for disseminating research findings are journals. Studies suggest that the recognition of journals and the relevance of articles are correlated. Within our dataset, there were 904 publications related to WBE that were published in 188 journals, out of which 36 matched the requirements (minimum number of documents per source: 5). In terms of sources of highly cited articles and the total number of relevant documents published, *Science of the Total Environment* occupied the leading spot (Figure 5a). Furthermore, there are substantial differences and influence gaps between this journal and the lower-ranked journals. *Science of the Total Environment* has published 245 articles; *Water Research*, the second most active journal, has published 56 articles, followed by *Environmental Research* with 48 articles. Similarly, the total number of citations from these three journals was 8170, 2689, and 709, respectively (Figure 5b). Journals' connections and relationships within the broader scientific landscape are of extreme importance. Apart from the total number of publications and total number of citations, other important journal parameters, including citations per publication, SJRs (scientific journal rankings), SNIP (source normalized impact per paper), cite score, quartiles, publisher, and publishing

countries of the top ten influential journals, can be found in Supplementary Table S2. It is noteworthy that these parameters are useful for gauging a journal's impact, but they have their own limitations and should be used in combination with other considerations while evaluating the quality and significance of academic journals [82].

**Table 4.** Top 10 highly cited articles related to the WBE of SARS-CoV-2 and respiratory viruses as of 6 July 2023.

| Rank | Authors | Article Title | Journal | Citation |
|---|---|---|---|---|
| 1 | Sojobi et al., 2022 [75] | Impact of sewer overflow on public health: A comprehensive scientometric analysis and systematic review | *Environmental Research* | 479 |
| 2 | Castro et al., 2022 [76] | Global occurrence of SARS-CoV-2 in environmental aquatic matrices and its implications for sanitation and vulnerabilities in Brazil and developing countries | *International Journal of Environmental Health Research* | 354 |
| 3 | Robins et al., 2022 [79] | Research needs for optimising wastewater-based epidemiology monitoring for public health protection | *Journal of Water and Health* | 324 |
| 4 | Abdeldayem et al., 2022 [77] | Viral outbreaks detection and surveillance using wastewater-based epidemiology, viral air sampling, and machine learning techniques: A comprehensive review and outlook | *Science of the Total Environment* | 310 |
| 5 | Donia et al., 2021 [82] | COVID-19 Crisis Creates Opportunity towards Global Monitoring & Surveillance | *Pathogens* | 305 |
| 6 | Bhalla et al., 2020 [80] | Opportunities and Challenges for Biosensors and Nanoscale Analytical Tools for Pandemics: COVID-19 | *Acs Nano* | 305 |
| 7 | Buonerba et al., 2021 [78] | Coronavirus in water media: Analysis, fate, disinfection and epidemiological applications | *Journal of Hazardous Materials* | 245 |
| 8 | Maryam et al., 2023 [14] | COVID-19 surveillance in wastewater: An epidemiological tool for the monitoring of SARS-CoV-2 | *Frontiers In Cellular and Infection Microbiology* | 242 |
| 9 | Corpuz et al., 2020 [17] | Viruses in wastewater: occurrence, abundance and detection methods | *Science of the Total Environment* | 242 |
| 10 | Mohapatra et al., 2021 [83] | The novel SARS-CoV-2 pandemic: Possible environmental transmission, detection, persistence and fate during wastewater and water treatment | *Science of the Total Environment* | 240 |

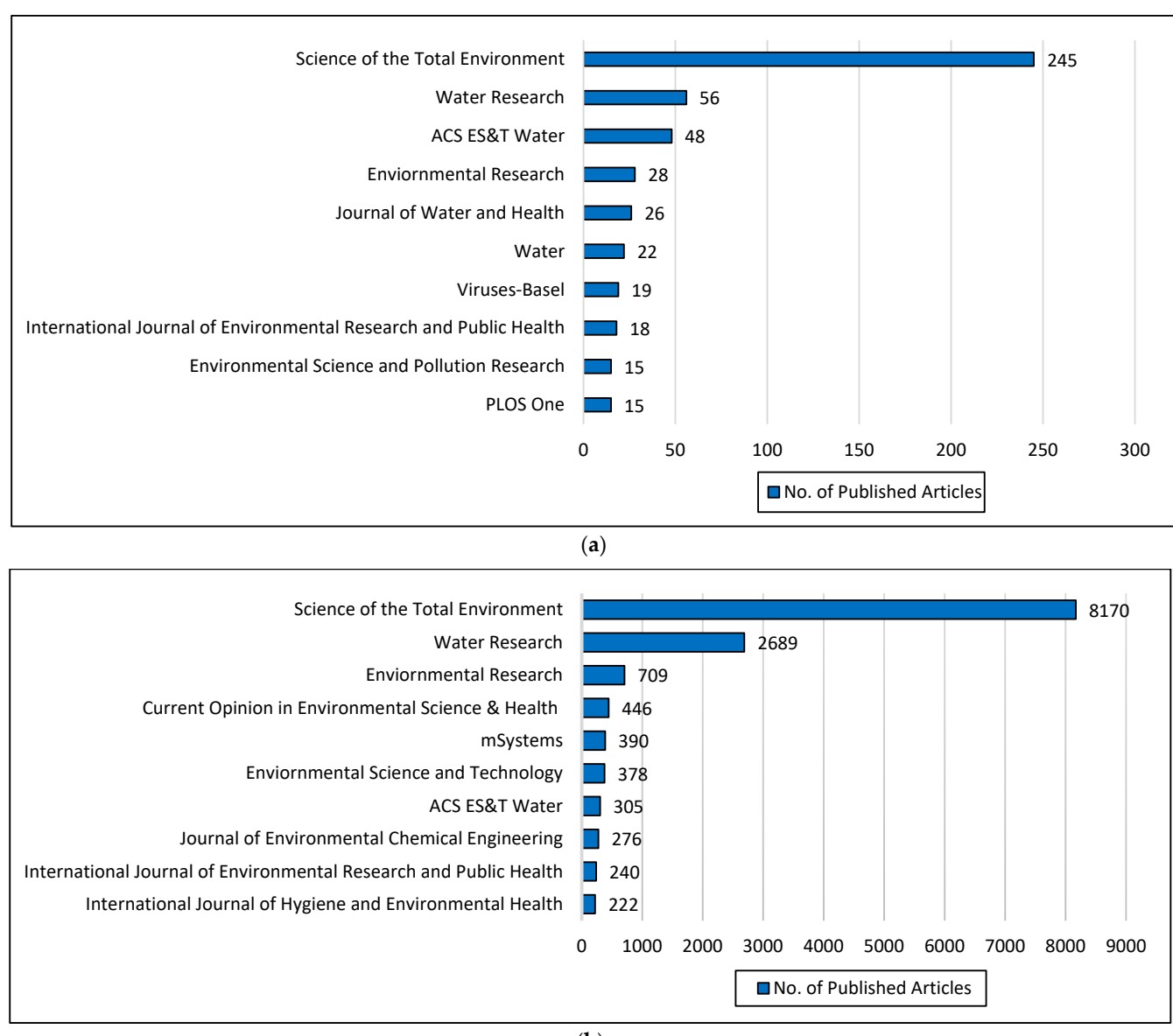

**Figure 5.** (**a**) Top productive journals in terms of the published literature related to the WBE of SARS-CoV-2 and respiratory viruses. (**b**) The most significant journals in terms of the number of citations to their articles related to the WBE of SARS-CoV-2 and respiratory viruses.

## 4. Research Gaps and Future Prospects

Our study identified the research gaps pertaining to virus concentration methods, analytical methods, and population biomarkers for RNA normalization in wastewater. These areas are important to decipher because of the variability among the techniques employed across different studies. Virus concentration methods can impact the efficiency of capturing viral particles from wastewater, which, in turn, is crucial for accurate detection and quantification.

The choice of analytical technique can impact the relationship between wastewater signals and clinical cases. RT-qPCR and digital PCR are available and widely used for the detection of respiratory viruses in wastewater, but further research is required to effectively pre-process and recover viral RNA and optimize RNA extraction from samples of different wastewater types. The major disadvantages of both of the above-mentioned methods are their cost-ineffectiveness and high time consumption. It is, therefore, essential that future research focus on the development of field-deployable, sensitive, and cost-effective

techniques/methods so that WBE can be practically implemented in underdeveloped or low-income countries.

Future research in field-deployable point-of-care devices will support mitigating the higher costs associated with WBE as it reduces the necessity of sample transportation to centralized molecular laboratories. Automation and high throughput are a few of the key desirable features in such devices that are still in developing phases and require further optimization and development. The need for an on-site viral RNA assessment has already been realized by the scientific community. The development of a biosensor-based dipstick kind of test for WBE analysis would be ideal [83]. The full-scale development and application of such a biosensor-based test would help ensure the equal inclusion of low-income countries in the field of WBE.

Additionally, another important direction in which WBE research is heading is assessing the relative effectiveness of different normalization methods. As discussed earlier, different normalization approaches have been employed by scientists to address the variables affecting viral RNA concentration during WBE analyses. Population biomarkers for RNA normalization can influence viral RNA quantification due to variations in wastewater flow and composition. As each geographical location has a specific set of physicochemical, environmental, and biological conditions, an agreement on a universal normalization technique warrants further investigation. Future research on WBE is expected to address the problem of different normalization techniques by comparing and even employing, in combination, different normalization strategies for different circumstances. However, the influence of the selected normalization technique on its relationship with clinical cases is an important area that is yet to be thoroughly explored. The comprehensive examination performed on the research gaps and future research directions would not only foster harmonization among studies but also enhance the potential for this approach to serve as a valuable tool in aiding the standardization of protocols.

The processes of WBE, as with other evolving processes, are also susceptible to uncertainties, non-reliabilities, and biases and, in many scenarios, can be impacted by several pre-analysis and post-analysis variables, such as sample collection, sample pre-treatments, RNA extraction, viral load analysis, changes in environmental conditions, and the framework and design of wastewater pathways from sewers to WWTP [84,85]. After the massive trend in research and application of WBE for SARS-CoV-2 and other viruses, the scientific community is unanimously calling on policymakers and decision-makers to establish a framework for standardized procedures for the surveillance and monitoring of wastewater so that patchwork being carried out on different levels can be consolidated.

International cooperation is of extreme importance in addressing most of the challenges tied to variations in wastewater composition and environmental conditions. Knowledge exchange and resource sharing between different countries enhance overall scientific expertise and lead to adaptable solutions. Through collaboration, standardized methods and protocols can be achieved, leading to meaningful data comparison. A standardized universal protocol would consolidate the science of methodology and enhance the ease of inter-country data sharing. Several documents have already been published by different research groups, suggesting and emphasizing the need for a comprehensive and standardized protocol for all the steps of WBE, from sample collection to data reporting and interpretations [85,86].

Moreover, by pooling data from different regions, a more comprehensive understanding of the behavior of viruses in wastewater can be achieved. There are several studies arguing about the efficacy of the phases of wastewater (solid vs. liquid) for the correct assessment of viral RNA. Similarly, for an accurate estimation of disease incidence, studies addressing the half-lives of many respiratory viruses in different wastewater matrices are also needed. The aggregated data can also lead to more accurate assessments of public health risks and trends, helping to guide effective response measures.

## 5. Conclusions

Our bibliometric analysis has provided an overall snapshot of the research landscape of the WBE of SARS-CoV-2 and other respiratory viruses. Herein, we analyzed a total of 904 publications that were published by authors from 87 countries over the last 4 years. Despite the lower incidence of reported clinical cases of COVID-19 in 2023, the number of publications on the WBE of SARS-CoV-2 remained consistent with previous years. The WBE of COVID-19 is still the most popular topic, though the scientific community has started exploring avenues of WBE for other respiratory viruses, particularly influenza. The high number of studies represents the value and significance of WBE and the important role it plays in the field of surveillance of SARS-CoV-2 and other respiratory viruses. Our study highlighted that method development, source and survival of the virus, development of WBE as a surveillance or early-warning detection system, and variants of concern were highly important topics.

The drive to enhance and implement advancements in the existing methodology is expected to persist in the coming years. WBE is increasingly expanding its reach toward new laboratories and extending its application to encompass a broader range of viruses. The introduction of field-deployable biosensors in the field of WBE is expected soon. Environmental engineers, microbiologists, and public health professionals can collaborate and provide a comprehensive and practical standardized protocol guide for WBE. Such multidisciplinary collaboration would be pivotal in advancing WBE as a reliable tool for monitoring and mitigating the spread of respiratory viruses in communities worldwide. It is anticipated that future studies in this domain will continue to focus on method development, including data normalization, fate, transmission, and survival of viruses, improvements to develop a smart early-warning virus detection system, and variants of concern.

**Supplementary Materials:** The following supporting information can be downloaded at https://www.mdpi.com/article/10.3390/w15193460/s1. Table S1: The top highly co-cited references in three largest clusters. Table S2: Citations per publication, SJR (scientific journal rankings), SNIP (source normalized impact per paper), cite score, quartiles, publisher, and publishing countries of top 10 influential Journals.

**Author Contributions:** Conceptualization, H.W. and J.A.; methodology, H.W.; software, R.A.; validation, H.W., J.A. and K.A.G.; formal analysis, H.W. and R.A.; investigation, H.W. and C.J.O.; resources, K.A.G.; data curation, R.A.; writing—original draft preparation, H.W.; writing—review and editing, K.A.G.; visualization, R.A.; supervision, K.A.G.; project administration, C.J.O.; funding acquisition, K.A.G. and C.J.O. All authors have read and agreed to the published version of the manuscript.

**Funding:** This research was funded by Health Canada under the COVID-19 Safe Restart Agreement (SRA) Program.

**Data Availability Statement:** The data supporting the findings of this study is available within the article and its supplementary materials.

**Conflicts of Interest:** The authors declare no conflict of interest.

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
