# Peer review of "Wastewater-Based Epidemiology of SARS-CoV-2 and Other Respiratory Viruses: Bibliometric Tracking of the Last Decade and Emerging Research Directions"

_water, doi:10.3390/w15193460_

Round 1

Reviewer 1 Report

the manuscript can be accepted in present form.

Author Response

The authors are thankful to reviewer no. 1 for acknowledgement of our work.

Reviewer 2 Report

1- what is relationship betwwen coved-19 and wastewater?

2- what are type of sutable treatment process selected to treat this type of polluted weste water 

3- English should be improved throughout the manuscript.

4- Add a comparative study table based on the literature review

Thank you

Author Response

Dear Editor and Reviewers,

Firstly, all the authors are thankful for the suggestions by the editor and the reviewers. We believe that the comments have really helped us in improving the overall quality of the manuscript. We have carefully revised the manuscript and reduced the similarity index from 20% to 14%. However, the authors would like to point out that the reason for the relatively high similarity index was mostly the names of the authors and publication titles used in tables and text for highlighting the research direction trends. The relevancy of all the references is also double checked. The point-by-point response of the reviewer’s comments is as follows:

Reviewer No. 2

Point 1: What is relationship between coved-19 and wastewater?

Response 1: Genetic fragments of the virus that causes COVID-19 are excreted by humans and measurable in wastewater samples. Wastewater surveillance for SARS-CoV-2 is a complimentary tool for tracking the prevalence and spread of disease in a community. This is described on lines 39-57 of the manuscript.

Point 2: what are types of sutable treatment process selected to treat this type of polluted weste water?

Response 2: Different wastewater treatment processes are used to treat viruses in wastewater, including Activated Sludge, UV Disinfection, and Ozonation. However, the reviewer should note that our study is not focused on reviewing methods for treating viruses in wastewater, rather, we conduct a bibliometric analysis of the evolution of research topics related to wastewater-based epidemiology and various respiratory viruses, including SARS-CoV-2.  

Point 3: English should be improved throughout the manuscript.

Response 3: English of the manuscript has now been improved throughout the manuscript.

Point 4: Add a comparative study table based on the literature review.

Response 4: Thank you for the suggestion, which we gave careful consideration to. We have decided that since this not a conventional review article, rather a bibliometric analysis, a table summarizing our findings would not add more information to the paper than what the figures already provide. In fact, we believe that our visualizations are more effective than a table that essentially repeats content already provided in the text.

Reviewer 3 Report

The manuscript entitled "Wastewater Based Epidemiology of SARS-CoV-2 and Respiratory Viruses: Bibliometric Tracking of the Last Decade and Further Research Needs" delivers a comprehensive snapshot of the existing state of WBE research and sets the stage for future developments in the field. Its insights are valuable for researchers, policymakers, and public health professionals working to harness the potential of wastewater-based epidemiology in monitoring and mitigating the spread of infectious diseases. Nevertheless, some points require improvement. In the TITLE, a hyphen is needed: Wastewater-Based. Optionally, the authors could change for this "Wastewater-Based Epidemiology of SARS-CoV-2 and Respiratory Viruses: A Bibliometric Analysis of the Last Decade and Emerging Research Directions". The ABSTRACT outlines the paper's motivation but lacks clarity on specific aims and findings. In the INTRODUCTION, the authors should ponder streamlining sentences and eliminating redundant phrases. Emphasize the novel aspects of your study, such as its focus on respiratory viruses and the application of bibliometric analysis. The specific research questions your study seeks to answer should be clearly stated. In the MATERIALS AND METHODS section, the data sources and search strategies could be briefly made by avoiding lengthy explanations. For instance, consider summarizing the description of the Web of Science (WOS) and its utility more succinctly. In the RESULTS AND DISCUSSION section, the authors could condense the description of the type and distribution of articles. The formatting of the charts needs to be improved for a more effective presentation. In section 4, the authors could emphasize which research gaps are most urgent or critical to address. For example, the authors mentioned the need for cost-effective and field-deployable techniques for WBE, which could be highlighted as a top priority. Provide specific recommendations for future research in these areas; mention how international cooperation can help overcome challenges related to differences in wastewater composition and environmental conditions.

The authors should provide a forward-looking perspective on the field in the CONCLUSIONS section. Discuss the expected developments, such as introducing fieldable biosensors and the collaborative efforts of environmental engineers, microbiologists, and public health professionals to establish standardized protocols. The authors should proofread the manuscript for any typographical errors or inconsistencies.

The authors should proofread the manuscript for any typographical errors or inconsistencies.

Author Response

Reviewer No. 3

Point 1: In the TITLE, a hyphen is needed: Wastewater-Based. Optionally, the authors could change for this "Wastewater-Based Epidemiology of SARS-CoV-2 and Respiratory Viruses: A Bibliometric Analysis of the Last Decade and Emerging Research Directions". 

Response 1: A hyphen is added, and the title of the article is also updated to “Wastewater-Based Epidemiology of SARS-CoV-2 and Other Respiratory Viruses: Bibliometric Tracking of the Last Decade and Emerging Research Directions”. Thank you for the suggestion.

Point 2: The ABSTRACT outlines the paper's motivation but lacks clarity on specific aims and findings.

Response 2: The abstract has been significantly revised to include the specific aims and findings (Line 20-21; 27-29).

Point 3: In the INTRODUCTION, the authors should ponder streamlining sentences and eliminating redundant phrases. Emphasize the novel aspects of your study, such as its focus on respiratory viruses and the application of bibliometric analysis. The specific research questions your study seeks to answer should be clearly stated.

Response 3: The introduction section has been revised to make sentences clearer and more concise. Also, many redundant phrases have been removed. A few lines about the focus of WBE on other respiratory viruses have been added (Line 72-75). The specific research questions of our study are now slightly revised and clearly stated (Line 99-106).

Point 4: In the MATERIALS AND METHODS section, the data sources and search strategies could be briefly made by avoiding lengthy explanations. For instance, consider summarizing the description of the Web of Science (WOS) and its utility more succinctly.

Response 4: The relevant section has now been significantly condensed from 571 to 138 words (Line 120-129).

Point 5: In the RESULTS AND DISCUSSION section, the authors could condense the description of the type and distribution of articles. The formatting of the charts needs to be improved for a more effective presentation.

Response 5: The description in relevant section has been condensed from 285 words to 232 words (Line 155-172). We have also reformatted Figure 1, Figure 2b, Figure 5a, and Figure 5b for effective presentation. Specifically, Figure one is replotted with contrasting colors and thick out lines and more visible line graph. Figure 2b is also replotted excluding two most common terms to make density plot more unbiased. In Figure 5a and Figure 5B more prominent colors and outlines are now used.

Point 6: In section 4, the authors could emphasize which research gaps are most urgent or critical to address. For example, the authors mentioned the need for cost-effective and field-deployable techniques for WBE, which could be highlighted as a top priority. Provide specific recommendations for future research in these areas; mention how international cooperation can help overcome challenges related to differences in wastewater composition and environmental conditions.

Response 6: Section 4 has been significantly revised to address the reviewers’ concerns (Line 454-487). The last two paragraphs are rewritten with an addition of a new paragraph about utility of international cooperation in WBE (Line 488-504).

Point 7: The authors should provide a forward-looking perspective on the field in the CONCLUSIONS section. Discuss the expected developments, such as introducing fieldable biosensors and the collaborative efforts of environmental engineers, microbiologists, and public health professionals to establish standardized protocols.

Response 7: The second paragraph of the conclusion section has been revised and additional content has been added as per authors suggestion (Line 519-529).

Point 8: The authors should proofread the manuscript for any typographical errors or inconsistencies.

Response 8: The whole manuscript has been thoroughly proofread for any typographical errors and inconsistencies.